# Mitral Valve Surgery via Upper Ministernotomy: Single-Centre Experience in More than 400 Patients

**DOI:** 10.3390/medicina57111179

**Published:** 2021-10-30

**Authors:** Medhat Radwan, Christoph Salewski, Florian Hecker, Aleksandra Miskovic, Petar Risteski, Jan Hlavicka, Anton Moritz, Thomas Walther, Tomas Holubec

**Affiliations:** 1Division of Thoracic and Cardiovascular Surgery, University of Tübingen, 72076 Tübingen, Germany; medhat.radwan@med.uni-tuebingen.de (M.R.); christoph.salewski@uni-tuebingen.de (C.S.); Petar.Risteski@med.uni-tuebingen.de (P.R.); 2Department of Cardiovascular Surgery, University Hospital Frankfurt, Johann Wolfgang Goethe University Frankfurt, 60590 Frankfurt am Main, Germany; florian.hecker@kgu.de (F.H.); Aleksandra.Miskovic@kgu.de (A.M.); Jan.Hlavicka@kgu.de (J.H.); moritzanton@web.de (A.M.); Thomas.Walther@kgu.de (T.W.)

**Keywords:** minimally invasive surgery, mitral valve surgery, partial upper sternotomy

## Abstract

*Background:* Minimally invasive mitral valve (MV) surgery has emerged as an alternative to conventional sternotomy aiming to decrease surgical trauma. The aim of the study was to describe our experience with minimally invasive MV surgery through partial upper sternotomy (PUS) regarding short- and long-term outcomes. *Methods:* From January 2004 through March 2014, 419 patients with a median age of 58.9 years (interquartile range 18.7; 31.7% females) underwent isolated primary MV surgery using PUS. Myxomatous degenerative MV disease was the predominant pathology (77%). The patients’ mean EuroSCORE II risk profile was 3.9 ± 3.6%. *Results:* Mitral valve repair was performed in 384 patients (91.6%) and replacement in 35 patients (8.4%). Thirty-day mortality was 3.1%. In total, 29 (6.9%) deaths occurred during the follow-up. The overall estimated survival at 1, 5, and 10 years was 93.1 ± 1.3%, 87.1 ± 1.9%, and 81.1 ± 3.4%. Reoperation was necessary in 14 (3.3%) patients. The overall freedom from MV reoperation at 1, 5, and 10 years was 98.2 ± 0.7%, 96.1 ± 1.2%, and 86.7 ± 6.7% and the overall freedom from recurrent MV regurgitation > grade 2 in repaired valves at 1, 5, and 10 years was 98.8 ± 0.6%, 98.8 ± 0.6%, and 94.6 ± 3.3%. *Conclusions:* Minimally invasive MV surgery via PUS can be performed with particularly good early and late results. Thus, the PUS approach with the use of standard surgical instruments and cannulation techniques can be a valuable option for the MV surgery either in patients contraindicated or not suitable to minithoracotomy.

## 1. Introduction

Minimally invasive cardiac surgery continues to grow in popularity owing to improvements in surgical technique and technology, together with the acceptance of minimally invasive approaches for operations previously performed through a conventional median sternotomy [1,2,3,4]. A variety of minimally invasive approaches to the mitral valve (MV) have been developed in the early 1990s aiming to decrease the surgical trauma by minimizing the size of incisions and modifying the approach to the MV by avoiding a median sternotomy. These approaches include partial sternotomy (ministernotomy), parasternal incisions, minithoracotomy, and total endoscopic/robotic access [5,6,7,8,9]. The main goal is to achieve identical technical perfection with the advantages of a limited skin incision. The reported benefits include a reduced amount of blood transfusion, preserved pulmonary function, better chest stability, less postoperative pain, improved postoperative recovery, decreased hospital length of stay, and ultimately faster return to all activities of daily living [10,11,12,13,14].

In the late 1990s, we adopted a minimally invasive approach to valve surgery via partial upper sternotomy (PUS) which became a standard access in our institution for aortic, mitral, and multiple valve surgery, as well as for arch surgery over the years [15,16,17].

The aim of this study was to analyze the short- and long-term outcomes of isolated minimally invasive MV surgery through PUS with regard to survival, MV-related reoperation, and recurrent MV regurgitation.

## 2. Materials and Methods

### 2.1. Study Population and Clinical Data

The study was approved by the local Ethical Committee of University Hospital Frankfurt, and an informed consent was obtained from each patient.

Four-hundred-nineteen patients with the median age of 58.9 (interquartile range 18.7) years (31.7% females), who underwent minimally invasive MV surgery via PUS at the Department of Cardiovascular Surgery, University Hospital Frankfurt from January 2004 through March 2014, were included in the study. Children and adult patients with concomitant valve, coronary, or aortic surgery were excluded from the study (Table 1).

Before surgery, all patients were routinely examined by transthoracic echocardiography (TTE), coronary angiography, and/or cardiac computed tomography (CT). An intraoperative transoesophageal echocardiography (TEE) was performed in all patients to evaluate valve function pre- and post-cardiopulmonary bypass to facilitate the surgical procedure and to assess the immediate result of the MV surgery as well as the left ventricular function during weaning from the cardio-pulmonary bypass and adequacy of de-airing of the cardiac chambers. The echocardiographic classification of residual/recurrent MV regurgitation was as follows: grade 0—none/trace; grade 1—mild; grade 2—moderate; grade 3—moderately severe; and grade 4—severe.

The postoperative anticoagulation regimen included initial subcutaneous low-molecular weight heparin for the first days and in parallel oral anticoagulation with vitamin K antagonist. Oral anticoagulation was continued for 6 weeks after an uncomplicated repair and adapted to the needs of atrial fibrillation or implanted prostheses thereafter. All patients received a TTE at discharge.

All patients were followed-up prospectively and systematically by means of annually mailed questionnaires or phone interviews and/or by clinical assessment and TTE in our outpatient clinic. For patients not seen personally we retrieved the clinical assessment and echocardiography reports from the attending cardiologist.

The completeness of follow-up was calculated as the portion of the actual observed patient years divided by the maximum of observable patient years as described by Akins et al. [18]. The entry to follow-up was the date of the operation. Follow-up was calculated until: 1. knowledge of death, 2. lost to follow-up, 3. last census date (either 1 January 2015 query at local authorities or clinical follow-up within 2015), and 4. end of study on 31 December 2015. The clinical follow-up time C was 87%. The median follow-up was 5.5 years (range 0–11). The total follow-up was 2366 patient-years. The echocardiographic follow-up was available in 96% (401/419) of patients. This accounts for 47% of the study period. The median echocardiographic follow-up was 2.9 years (range 0–11).

### 2.2. Surgical Technique

The operative technique has been previously described in detail [15,16]. Briefly, after cross-clamping the aorta and arresting the heart using antegrade cardioplegia the MV was approached through a superior trans-septal atriotomy, in the majority of cases with extension to the left atrial roof. Several stay sutures were placed on the interatrial septum, the base of the left atrial appendage, and the mural portion of the mitral annulus to expose the MV. Exposure was further optimized after placing the annuloplasty stitches through both trigoni fibrosi. After a proper valve analysis, either an MV repair or eventually a valve replacement with mechanical or biological prosthesis was performed.

### 2.3. Statistical Analysis

Valve-related outcomes were defined according to the published guidelines [18]. Continuous and discrete variables were expressed as mean ± SD or median and interquartile range (IQR) for data not normally distributed. Categorical and ordinal variables were expressed by number and percentage of observations. Continuous and discrete variables were compared using a two-sample t-test or Mann–Whitney test, where appropriate. The probability of freedom from event was calculated according to the Kaplan–Meier method. Survival and freedom-from-event curves were compared by log-rank test. A *p*-value < 0.05 was considered to indicate statistical significance. A statistical analysis was performed using Stat view (Carry, NC, USA) and IBM SPSS (version 25 for MS Windows; IBM Corporation, Armonk, NY, USA).

## 3. Results

### 3.1. Operative Data

Mitral valve repair was performed in 384 patients (91.6%) and replacement in 35 patients (8.4%). Mainly used MV repair techniques were triangular or quadrangular resection followed by artificial chord implantation accomplished by ring annuloplasty. The majority of the patients (86%) received an open flexible ring (Cosgrove-Edwards ring). A more detailed description of repair techniques and other perioperative characteristics are provided in Table 2. A second pump run to improve the repair or to replace the valve was needed in 15 patients (3.6%). The MV was replaced in 20 cases (4.8%) using a mechanical prosthesis and in 15 cases (3.6%) using a biological prosthesis (Table 2).

### 3.2. Early Postoperative Outcomes

Thirty-day mortality occurred in 13 cases (3.1%). In eight of these patients (61%), a re-exploration due to postoperative bleeding was necessary. The cause of death was multiple organ failure in six, sepsis in three, low cardiac output syndrome in one, major stroke in one, and other causes in two patients, respectively. Other early postoperative data are shown in detail in Table 3. The TTE data at discharge are presented in Table 4.

### 3.3. Survival and Late Clinical Outcomes

During the follow-up, 26 (6.2%) patients died. The overall estimated survival at 1, 5, and 10 years was 93.1 ± 1.3%, 87.1 ± 1.9%, and 81.1 ± 3.4% (Figure 1). The Kaplan–Meier survival estimator for the overall cohort was 9.6 ± 0.2 years (CI 9.197–9.986). After dividing the overall cohort into repair versus replacement group, the survival at 5- and 10-years was 89.3 ± 1.8% and 82.1 ± 4.1% vs. 53.8 ± 12.1% and 53.8 ± 12.1% (*p* < 0.001), respectively.

Fourteen (3.3%) patients required reoperation after MV repair during the follow-up interval due to the following reasons: annulus re-dilatation and/or recurrent leaflet prolapse in 11 and endocarditis with leaflet perforation in three patients. In four patients, re-repair of the MV was successfully performed. All reoperations were performed through the full median sternotomy. The detailed data of all reoperated patients including the exact reason of reoperation are shown in Table 5. The overall estimated freedom from MV reoperation at 1, 5, and 10 years was 98.2 ± 0.7%, 96.1 ± 1.2%, and 86.7 ± 6.7% (Figure 2). The Kaplan–Meier estimator for freedom of reoperation for the overall cohort was 10.4 ± 2.2 years (CI 9.940–10.823).

The overall estimated freedom from recurrent MV regurgitation >grade 2 in repaired valves at 1, 5, and 10 years was 98.8 ± 0.6%, 98.8 ± 0.6%, and 94.6 ± 3.3% (Figure 3). The Kaplan–Meier estimator for freedom from mitral insufficiency > grade 2 was 10.8 ± 0.15 years (CI 10.453–11.053). No statistically significant difference was found in regard to freedom from reoperation (*p* = 0.95) or regurgitation > 2 (*p* = 0.479) after dividing the overall cohort into repair versus replacement groups.

The TTE data at the latest follow-up of all living and non-reoperated patients are shown in Table 4. The TTE data analysis between discharge and last follow-up was able to show a statistically significant increase in MR grade (0.21 to 0.61, *p* = 0.01) and decrease in MV orifice area (2.73 to 2.44 cm^2^, *p* = 0.01).

## 4. Discussion

Mitral valve surgery including repair or replacement, especially in complex pathologies, can be a challenging cardio-surgical procedure. Despite this fact, many surgeons worldwide perform MV surgery using a minimally invasive approach. This trend is supported by currently available evidence. Cheng and colleagues suggested minimally invasive MV surgery to be associated with decreased bleeding, blood transfusion, atrial fibrillation, sternal wound infections, ventilation time, intensive care unit stay, and length of hospital stay, and more expeditious return to the normal physical activity, whereas no differences in survival and valve re-intervention in the long-term were found [19].

As a surgical technique, two different minimally invasive access modalities to the MV are currently available, accepted, and under widespread clinical use: 1. right lateral minithoracotomy (direct-vision, video-assisted, total endoscopic, and robotic) with usually peripheral cannulation; and 2. PUS with central cannulation and a trans-septal approach to the MV. For the surgeon who decides to use a minimally invasive approach for MV operation a decision has to be made mainly based on the type of incision and the way of cannulation. Whereas right lateral minithoracotomy MV surgery usually is performed together with retrograde femoral perfusion, the PUS technique can be applied together with standard antegrade aortic perfusion. Thus, it is a potentially valid therapeutic approach whenever retrograde femoral perfusion should be avoided (e.g., in severe peripheral arterial occlusive disease, severe plaques in the aorta, etc.), and is quite similar for patients where right lateral minithoracotomy should be avoided (e.g., previous surgery in this field leading to adhesions, severe lung disease, etc.).

In the past years, PUS has become an attractive and frequently used approach for minimally invasive aortic valve replacement [20]. Moreover, the indications for this minimally invasive approach were extended to even more complex operations, which can be safely performed via PUS [16,17,21,22,23]. In our department, we expanded the indication for this minimally invasive access and have been using it as a standard surgical approach to the MV even with concomitant tricuspid and/or aortic valve surgery starting in 1998. The PUS access may be performed to the left or right fourth intercostal spaces. In our practice, we performed it in the left fourth intercostal space. The advantages of extending the PUS to the left fourth intercostal space are as follows: 1. the heart lies in the thorax more to the left than to the right of the midline, 2. the MV is more easily accessible using the left instead of the right extension of the PUS, and 3. the surgeon does not have to work under the sternum, as is necessary when the PUS ends up in the right intercostal space.

The current study shows very good results of 419 minimally invasive MV procedures via PUS in an all-comers cohort with a 30-day mortality of 3.1%. Seeburger and colleagues reported outcomes in a series of 1.536 consecutive patients who underwent minimally invasive MV surgery for regurgitation through a right lateral minithoracotomy with a 30-day mortality of 2.4% [24]. Due to an all-comers population in our study, the 30-day mortality of 3.1% seems elevated, but with a mean EuroSCORE II of 3.9 ± 3.6% expected, the observed mortality seems to be favorable. The overall survival at 5 and 10 years was 87.1 ± 1.9% and 81.1 ± 3.4%, respectively. Patients undergoing MV replacement had a higher risk of mortality in the follow-up (*p* < 0.001), which is probably due to the fact that these patients had a higher risk profile, rather than to the procedure itself. The 5-year survival of our patients receiving repair only of 89.3 ± 1.8% was even better compared to Seeburger’s series of 82.6% [24].

The freedom from MV-related reoperation and freedom from recurrent MV mitral insufficiency > grade 2 at 5 years: 96.1 ± 1.6% and 98.8 ± 0.6% and at 10 years: 86.7 ± 6.7% and 94.6 ± 3.3% respectively, were encouraging. The freedom from reoperation in Seeburger´s series was quite similar with 96.3% [24]. In a meta-analysis performed by Sündermann and colleagues published in 2014 including more than 20,000 patients from 45 studies, a perioperative morbidity following MV surgery has been reported [25]. The incidence of major neurological events following MV surgery via right antero-lateral minithoracotomy was 1.7%, which is comparable to our results with an incidence of major neurological events of 2%. The median of chest tube drainage during ICU stay in our cohort was 450 mL, which is also in line with this study.

The analysis comparing TTE data at discharge and at last follow-up (Table 4) revealed a solid performance of MV surgery via PUS approach; however, a slightly increasing MR grade (0.21 to 0.61, *p* = 0.01) over the follow-up period was observed. One of the reasons for the increase could be that the majority of the patients received an open flexible ring (Cosgrove-Edwards ring). Annular re-dilatation with this ring has been described and is possible, thus it could lead to recurrence of the MR. Additionally, in two of the reoperated patients an annular re-dilatation was observed and both patients were initially implanted with a Cosgrove-Edwards ring. Another finding was a slightly decreasing MV orifice area (2.73 to 2.44 cm^2^, *p* = 0.01). Due to this fact, one can postulate that this is not the acute effect of annuloplasty with strong ring downsizing, but rather the effect of MV disease evolution and progression after MV repair with leaflet repair and annuloplasty sutures.

Despite the stimulating results and some potential benefits of the PUS approach using transseptal access to the MV, there is definitely at least one major drawback. The increased rate of postoperative pacemaker implantation in nearly 5% of patients was slightly higher than in the literature of the minithoracotomy approach. However, it is comparable to other studies using the PUS access, as Oezpeker and colleagues reported a postoperative permanent pacemaker implantation rate of 4% in 103 patients [26]. The reported incidence of postoperative AV-block and subsequent pacemaker implantation can probably be explained by the possible injury of the conduction system through the transseptal access to the MV. Due to this drawback and because of the technically very demanding access to the MV via the interatrial groove in the PUS approach, some other groups considered different access to the MV. Little and colleagues proposed the left atrial dome incision as an optional access to the MV in the PUS approach [27]. They described their initial experience with this technique and concluded that adequate visualization of the MV for standard repair or replacement can be achieved. Additionally, this approach rarely divides the sinus node artery and therefore the rate of postoperative AV-block remains significantly lower than in the transseptal access. There are several limitations of this study. This was a retrospective, single-centre, non-randomized, observational study, and all inherent disadvantages apply. Additionally, the inability to determine the late causes of death and a lack of complete follow-up could be limiting.

## 5. Conclusions

This study provides short- and long-term results of minimally invasive MV surgery via PUS in a large all-comers cohort of patients. The results indicate that it can be performed safely with very good early and late results in both MV repair and replacement. In fact, the minimally invasive MV surgery via right antero-lateral minithoracotomy and peripheral cannulation is currently becoming more popular and has been adopted in the majority of centers, including our department. However, this approach demands a completely different setting, equipment, instruments, and especially thinking of the operating team. We therefore believe that the PUS approach with the use of standard surgical instruments and cannulation techniques can be a valuable alternative option for the MV surgery either in patients contraindicated or not suitable for minithoracotomy or in centers not willing or not able to establish the latter. In addition, the PUS access can even be easily extended to be used in patients requiring multiple valve surgery.

## Figures and Tables

**Figure 1 medicina-57-01179-f001:**
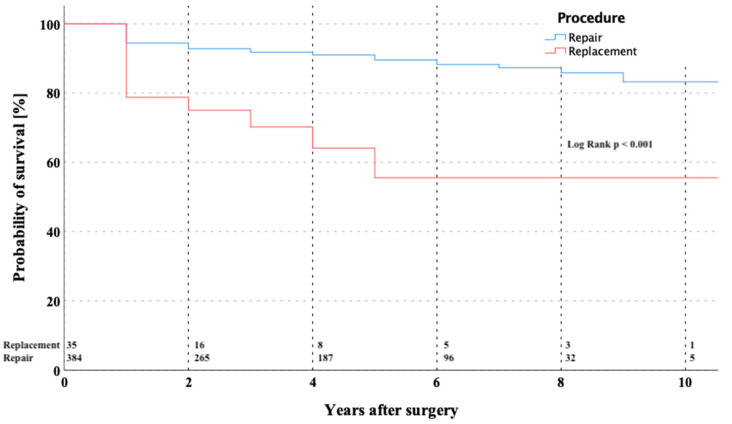
Kaplan–Meier curve showing survival in patients after mitral valve repair versus replacement through partial upper sternotomy.

**Figure 2 medicina-57-01179-f002:**
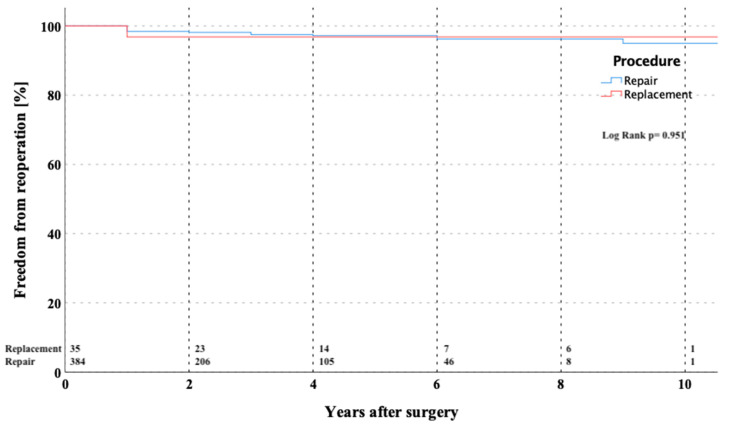
Kaplan–Meier curve showing freedom from MV-related reoperation in patients after MV repair versus replacement through partial upper sternotomy. MV—mitral valve.

**Figure 3 medicina-57-01179-f003:**
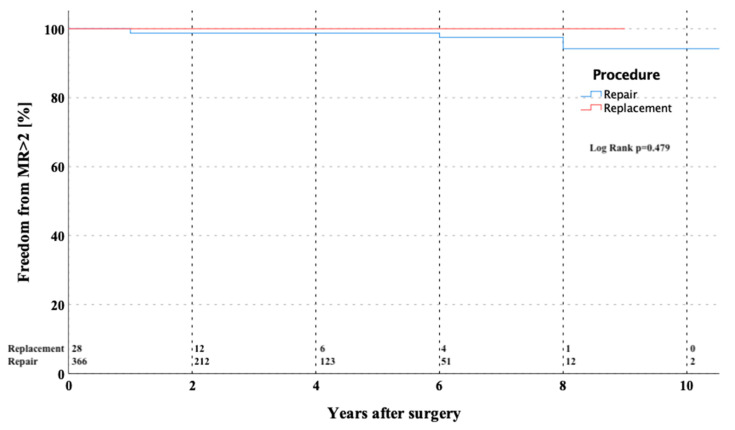
Kaplan–Meier curve showing freedom from MR > grade 2 in patients after MV repair versus replacement through partial upper sternotomy. MR—mitral valve regurgitation; MV—mitral valve.

**Table 1 medicina-57-01179-t001:** Preoperative characteristics.

Variable	Median/Number/Mean	%/IQR/SD
Age [years]	58.9	18.7
Female	133	31.7
Hypertension	326	53.3
Diabetes	34	8.1
Peripheral vascular disease	8	1.9
Previous stroke	27	6.4
EuroSCORE II	3.9	±3.6
Sinus rhythm	302	72.1
Pulmonary hypertension	147	35.1
COPD	38	9.1
Etiology:		
Myxomatous degenerative	323	77.1
Posterior leaflet prolapse	257	79.6
Bileaflet prolapse	31	9.6
Anterior leaflet prolapse	26	8.0
Commissural prolapse	9	2.8
Rheumatic	22	5.3
Infective	32	7.6
Isolated mitral ring dilatation	30	7.2
Other	12	2.9
Indication:		
Elective	340	80.7
Urgent	75	17.9
Emergency	4	1.0

COPD—chronic obstructive pulmonary disease; IQR—interquartile range; SD—standard deviation.

**Table 2 medicina-57-01179-t002:** Operative data.

Variable	Number/Median	%/IQR
Mitral valve repair	380	90.7
Triangular or quadrangular resection	276	65.8
Artificial chord implantation	139	33
Ring annuloplasty	380	100
Cosgrove-Edwards ring (flexible)	328	86.4
Carpentier-Edwards physio ring (semirigid)	47	12.3
Profile 3D ring (rigid)	5	1
Mitral valve replacement	39	9.3
Mechanical prosthesis	17	4.1
Biological prothesis	8	1.9
Left atrial ablation	61	14.5
Amputation or occlusion of left atrial appendage	261	62.2
Aortic cross-clamp time [min]	97.5	11.2
Cardio-pulmonary bypass time [min]	144.3	40.0

IQR—interquartile range; SD—standard deviation.

**Table 3 medicina-57-01179-t003:** Early postoperative data.

Variable	Number (%)/Median (IQR)
Reoperation for bleeding	31 (7.3%)
Blood loss during ICU stay [ml]	450/363
Myocardial infarction	2 (0.4%)
Postoperative dialysis	23 (5.4%)
Major neurological complications	8 (2%)
Gastrointestinal bleeding/ischemia	11 (2.6%)
Superficial sternal wound infection	16 (3.8%)
Deep sternal wound infection	4 (1%)
Postoperative pacemaker implantation	24 (5.7%)
30-day mortality	13 (3.1%)

**Table 4 medicina-57-01179-t004:** Echocardiographic data at discharge and follow-up.

Variable	At Discharge (Mean ± SD)	Last Follow-Up (Mean ± SD)	*p*-Value
Mitral regurgitation [grade]	0.21 ± 0.40	0.61 ± 0.66	0.01
LVEDD [mm]	51.84 ± 7.62	51.78 ± 8.79	0.92
LVESD [mm]	36.13 ± 8.15	35.39 ± 9.03	0.12
MV Pmean [mmHg]	4.02 ± 2.85	4.17 ± 2.89	0.51
MVA [cm^2^]	2.73 ± 1.05	2.44 ± 1.13	0.01
Left atrial diameter [mm]	45.44 ± 8.59	45.99 ± 8.60	0.68
LVEF [%]	55.4 ± 9.3	53.7 ± 8.8	0.23

LVEF—left ventricular ejection fraction; LVEDD—left ventricular end-diastolic diameter; LVESD—left ventricular end-systolic diameter; MV Pmean—mitral valve mean pressure gradient; MVA—mitral valve orifice area; SD—standard deviation.

**Table 5 medicina-57-01179-t005:** Data of reoperated patients due to mitral valve disease.

Patient	Age	Incidation to Operation	Operation	Time to Reoperation (Years)	Indication to Reoperation	Reoperationa
1	46	Myxomatous degenerative	MV repair	0.01	MV stenosis	MV replacement
2	41	Myxomatous degenerative	MV repair	0.03	MV stenosis	MV replacement
3	46	Myxomatous degenerative	MV replacement	0.09	MV regurgitation due to paravulvar leak	MV re-replacement
4	71	Myxomatous degenerative	MV repair	0.2	MV regurgitation due to endocarditis	MV replacement
5	48	Myxomatous degenerative	MV repair	0.5	MV regurgitation due to leaflet re-prolapse	MV re-repair
6	44	Rheumatic	MV replacement	0.6	MV regurgitation due to paravulvar leak	MV re-replacement
7	65	Myxomatous degenerative	MV repair	0.8	MV regurgitation due to leaflet re-prolapse	MV replacement
8	68	Myxomatous degenerative	MV repair	1.3	MV stenosis	MV replacement
9	51	Myxomatous degenerative	MV repair	2	MV regurgitation due to annulus re-dilatation	MV re-repair
10	44	Acute endocarditis	MV repair	2.2	MV regurgitation due to re-ndocarditis	MV replacement
11	56	Degenerative/calcification	MV repair	3	MV regurgitation due to annulus re-dilatation	MV re-repair
12	49	Myxomatous degenerative	MV repair	6	MV stenosis	MV replacement
13	46	Myxomatous degenerative	MV repair	6	MV regurgitation due to leaflet re-prolapse	MV re-repair
14	51	Degenerative/calcification	MV repair	8	MV regurgitation due to endocarditis	MV replacement

MV—Mitral valve.

## Data Availability

No additional data included.

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
