# Peer review of "Mitral Valve Surgery via Upper Ministernotomy: Single-Centre Experience in More than 400 Patients"

_medicina, 2021, doi:10.3390/medicina57111179_

Round 1

Reviewer 1 Report

It would be of public interest to hear (in the discussion section) your comment (experience) on the left atrial dome approach in PUS instead of the transeptal incision (performed as a modification or refinement of the procedure by some surgeons), reported to produce lesser injury to the conduction system and/or chance of division of sinus node artery achievable without prolongation of the operative time. (an applicable reference, doi: https://doi.org/10.1016/j.athoracsur.2008.03.043)

Otherwise, well-done manuscript.

Author Response

Comment #1:

It would be of public interest to hear (in the discussion section) your comment (experience) on the left atrial dome approach in PUS instead of the transeptal incision (performed as a modification or refinement of the procedure by some surgeons), reported to produce lesser injury to the conduction system and/or chance of division of sinus node artery achievable without prolongation of the operative time. (an applicable reference, doi: https://doi.org/10.1016/j.athoracsur.2008.03.043)

Thanks a lot for your valuable comment. We took your comment into account and have accomplished the Discussion section with this paragraph:

Due to this drawback and because of the technically very demanding access to the MV via interatrial groove in the PUS approach, some other groups considered different access to the MV. Little and colleagues proposed the left atrial dome incision as an optional access to the MV in the PUS approach [27]. They described their initial experience with this technique and concluded that adequate visualisation of the MV for standard repair or replacement can be achieved. Additionally, this approach rarely divides the sinus node artery and therefore the rate of postoperative AV-block remains significantly lower, than in the than in the transseptal access.

The above provided paragraph has been adjunct into the discussion section.

Comment #2:

Otherwise, well-done manuscript.

Thank you very much.

No changes.

Reviewer 2 Report

  • In the group underwent mitral valve repair more data on preoperative mitral valve pathology are needful. Since most of valves repaired were probably of degenerative ethyology, number/percentage of prolapse location ( posterior, anterior or bileaflet prolapse) and/or patients with isolated annular dilatation shuold  be mentioned
  • What were the ring types implanted (rigid, semirigid or flexible, full or semiring ) and what were the ring sizes ? These data are imortant in analysis of reoperatonts due to to reccurent MR. 
  • How do you explain the re-dilatation of the anulus as the cause of reccurent MR since all patient had rings implanted ? Annuloplasty ring is more or less rigid device firmly stiched to the annulus and annular re-dilatation is not possible except in case of semirngs, flexible ring or ring dehiscence? Need to be clarified in the text.
  • How do you explain that total number of rings implanted (419) far excedes the total number of repairs perfomed (380), Table 2 ? Need to be clarified in text.
  • How do you explained the stenosis after the repair in degenerative mitral insufficiency ? Ring mismatch could be an explanation for early post-op stenosis but in one patient appeared 1,3 years after the surgery? Need to be further analysed in the text.
  • It is neccessary to present ejection fraction (EF) in echocardiographic follow-up data table.
  • What was the lenght of hospital stay ? 
  • What was ICU blood loss ?
  • The units of measure need to be placed aside of variable presented, table 1. For example Age(years) instead of just Age.
  • Long term survival need to be completed with survival analysis of long in repair and replacemt group.
  • English language to improve in sentence line 115 - 116

Author Response

Comment #1:

In the group underwent mitral valve repair more data on preoperative mitral valve pathology are needful. Since most of valves repaired were probably of degenerative ethyology, number/percentage of prolapse location (posterior, anterior or bileaflet prolapse) and/or patients with isolated annular dilatation should  be mentioned.

Thank you very much for valuable comment and question. The aetiology was as follow: myxomatous degenerative 323 patients (posterior leaflet prolapse 257 patients = 79.6%, bileaflet 31 patients = 9.6%, anterior leaflet prolapse 26 patients = 8.0% and commissural prolapse in 9 patients = 2.8%) and isolated mitral ring dilatation 30 patients.

This has been amended in the Table 1.

Comment #2:

What were the ring types implanted (rigid, semirigid or flexible, full or semiring ) and what were the ring sizes? These data are important in analysis of reoperations due to recurrent MR.

Thanks a lot for your question. Following rings were implanted: Cosgrove-Edwards ring (flexible) in 328 patients = 86.4%, Carpentier-Edwards Physio ring (semirigid) in 47 patients = 12.3% and Medtronic Profile 3D ring (rigid) in 5 patients = 1.3%. We have not originally entered the ring sizes in our database; therefore, we are unfortunately unable to provide this data. Sorry.

This data has been added to the Table 2.

Comment #3:

How do you explain the re-dilatation of the anulus as the cause of recurrent MR since all patient had rings implanted? Annuloplasty ring is more or less rigid device firmly stitched to the annulus and annular re-dilatation is not possible except in case of semirings, flexible ring or ring dehiscence? Need to be clarified in the text.

Thank you very much for this valuable comment. As we pointed out already above, majority of the patient in the repair group received open flexible ring (Cosgrove-Edwards ring); therefore, re-dilatation of the annulus is possible and has been described. This case happened only in two reoperated patients; however, both have implanted this flexible ring.

Corresponding sentence has been added to the discussion section.

Comment #3:

How do you explain that total number of rings implanted (419) far exceeds the total number of repairs performed (380), Table 2? Need to be clarified in text.

Excellent point, thank you! Sorry for our incorrectness, the number is false.

The number has been corrected to 380 patients in Table 2.

Comment #4:

How do you explain the stenosis after the repair in degenerative mitral insufficiency? Ring mismatch could be an explanation for early post-op stenosis but in one patient appeared 1,3 years after the surgery? Need to be further analysed in the text.

Thanks a lot for this valuable comment and question. As you can see in Table 4, we have provided MV orifice area (MVA) at discharge and at latest follow-up. We were able to see significant decrease of MVA at last FU (2.73 to 2.44 cm2, p = 0.01). Due to this fact, one can postulate that this is not the acute effect of annuloplasty and strong downsizing, but rather the effect of MV disease evolution/progression after MV repair (leaflet repair, annuloplasty sutures).

Corresponding sentence has been added to the discussion section.

Comment #5:

It is necessary to present ejection fraction (EF) in echocardiographic follow-up data table.

The ejection fraction data (at discharge and at last follow-up) were generated from the database.

This data was amended in Table 4.

Comment #6:

What was the length of hospital stay?

Thank you for your question; however, we cannot supply this data, because we do not have this data. On the other hand, especially the length of the hospital stay is very unspecific parameter, because it varies very much depending on the health care system (e.g. Europe vs USA).

Comment #6:

What was ICU blood loss?

The median of ICU blood loss was 450 ml (IQR 363 ml).

This variable has been accomplished in the Table 3.

Comment #7:

The units of measure need to be placed aside of variable presented, table 1. For example, Age (years) instead of just Age.

Thank you very much for your important comment. Absolutely, correct!

This has been considered and adapted in the Tables. Additionally, the continuous variables have been re-considered for normality and consequently presented as median and IQR throughout the whole manuscript.

Comment #7:

Long-term survival need to be completed with survival analysis of long in repair and replacement group.

Thank you for this comment. We have split the survival curves into repair and replacement and performed a log rank test to assess difference in survival of the groups. The replacement group had a significantly lower long-term survival (p<0.001) than the repair group. No relevant difference was found in respect to freedom from reoperation or regurgitation >2.

All three Figures have been replaced by new graphics and figure legends were adapted. Additionally, and correspondingly, the text in results and discussion section were adapted and accomplished.

Comment #8:

English language to improve in sentence line 115 – 116

We need specification here, as spelling and grammar seem to be correct. However, due to your comment we checked this section again and removed following sentences since they do not apply in this manuscript. “Categorical and ordinal variables were compared using Pearson’s Chi-squared test or Fischer’s exact test, where appropriate. Logistic regression analysis was used to determine the predictors of survival and in-hospital mortality.”

We have added the following sentence into the Statistical Analysis section: ”Survival and freedom-from-event curves were compared by log-rank test.”

Reviewer 3 Report

Very important study for our quotidian practice. This study going to bring us more security to operate the patients who goes to MV surgery with UPS. The results even a single center, non-randomized study and observational, were goods and inside of the international results regarding the literature review.

Please, review this part of text:

The current study shows particularly good results of 419 minimally invasive MV pro- 215 cedures via PUS in an all comers cohort. The overall survival, freedom from MV-related 216 reoperation and freedom from recurrent MV mitral insufficiency > grade 2 at 10-years 217 were encouraging at 81.1 ± 3.4%, 95.1 ± 1.6% and 94.6 ± 3.3%, respectively.

Thank You.

Bests Regards.

Author Response

Comment #1:

Very important study for our quotidian practice. This study going to bring us more security to operate the patients who goes to MV surgery with UPS. The results even a single centre, non-randomized study and observational, were goods and inside of the international results regarding the literature review.

Thank you so much for your great comment.

No changes.

Comment #2:

Please, review this part of text: The current study shows particularly good results of 419 minimally invasive MV procedures via PUS in an all-comers cohort. The overall survival, freedom from MV-related reoperation and freedom from recurrent MV mitral insufficiency > grade 2 at 10-years were encouraging at 81.1 ± 3.4%, 95.1 ± 1.6% and 94.6 ± 3.3%, respectively.

Thank you very much for your point. We have reviewed and corrected properly this part of the text.

The above provided text has been implemented in the Discussion section of the manuscript.